# Biotreatments Using Microbial Mixed Cultures with Crude Glycerol and Waste Pinewood as Carbon Sources: Influence of Application on the Durability of Recycled Concrete

**DOI:** 10.3390/ma15031181

**Published:** 2022-02-03

**Authors:** Lorena Serrano-González, Daniel Merino-Maldonado, Andrea Antolín-Rodríguez, Paulo C. Lemos, Alice S. Pereira, Paulina Faria, Andrés Juan-Valdés, Julia García-González, Julia Mª Morán-del Pozo

**Affiliations:** 1Department of Engineering and Agricultural Sciences, School of Agricultural and Forest Engineering, University of León, Av. De Portugal 41, 24071 Leon, Spain; dmerm@unileon.es (D.M.-M.); aantr@unileon.es (A.A.-R.); andres.juan@unileon.es (A.J.-V.); julia.garcia@unileon.es (J.G.-G.); julia.moran@unileon.es (J.M.M.-d.P.); 2Associated Laboratory for Green Chemistry-Chemistry and Technology Network (LAQV-REQUIMTE), Department of Chemistry, NOVA School of Science and Technology (FCT NOVA), NOVA University of Lisbon, 2829-516 Caparica, Portugal; paulo.lemos@fct.unl.pt; 3Applied Molecular Biosciences Unit (UCIBIO), Department of Chemistry, NOVA School of Science and Technology, Universidade NOVA de Lisboa, 2829-516 Caparica, Portugal; masp@fct.unl.pt; 4Associate Laboratory i4HB—Institute for Health and Bioeconomy, NOVA School of Science and Technology, Universidade NOVA de Lisboa, 2829-516 Caparica, Portugal; 5Civil Engineering Research and Innovation for Sustainability (CERIS), Department of Civil Engineering, NOVA School of Science and Technology (FCT NOVA), NOVA University of Lisbon, 2829-516 Caparica, Portugal; paulina.faria@fct.unl.pt

**Keywords:** construction and demolition waste (CDW), biopolymers, recycled concrete, surface treatment, PHA-producing MMC, waterproof

## Abstract

Two eco-friendly healing bioproducts generated from microbial mixed cultures (MMC) for the production of polyhydroxyalkanoates (PHA) were used as surface treatments, with two residual materials used as the substrates, namely crude glycerol and pinewood bio-oil. Their ability to improve the durability of concrete samples containing recycled aggregates was assessed. To determine this protective capacity, 180 samples were analyzed using different tests, such as water penetration under pressure, capillary absorption, freeze–thaw and water droplet absorption test. Three types of conditions were used: outdoor–indoor exposure, re-application of biopolymers and application in vertical exposure conditions. The results showed reductions of up to 50% in the water penetration test and a delay in the water droplet absorption test of up to 150 times relative to the reference. The surface application of these bioproducts significantly reduced the degree of water penetration in recycled concrete, increasing its useful lifespan and proving to be a promising treatment for protecting concrete surfaces.

## 1. Introduction

The global population growth over the past 100 years has resulted in rapid industrial and urban development. Considering the properties of concrete, namely its versatility, resistance and low cost, a large increase in the demand for this material has been occurring, making it the second most extensively used material worldwide, only behind water [1]. 

According to The International Energy Agency [2], approximately 25 GT of concrete is used each year globally. Considering that aggregates occupy approximately 60–80% of the concrete volume [3] and that the demand for aggregates is increasing annually by 2.3%, this value is expected to increase from 40 billion tons in 2015 to 47.5 billion tons by 2023 [4]. Consequently, in some territories there is already a shortage of natural aggregates (both gravel and sand) [5,6].

The environmental problems associated with the production of concrete are not limited to the necessary consumption of natural aggregates, but also the large volume of residues generated, as the construction industry generates 50% of the global waste [7]. Those values represent a challenge to waste management due to the shortages of waste land and the high costs associated with the necessary treatment before disposal [8,9], which is why part of this waste is sent to landfills, resulting in consequent environmental degradation [10]. Concrete also largely contributes to the emission of CO2 [11], a naturally occurring greenhouse gas responsible for about 5–8% of man-made emissions [12]. Therefore, the concrete industry has become the major source of environmental pollution, having a massive impact on the environment [13].

For these reasons, over the last two decades an increasing research effort has been devoted to evaluating the sustainability of the construction and demolition sectors in urban systems [14]. Such research is particularly urgent in Europe, as the amount of demolition concrete waste is expected to increase dramatically, since the majority of concrete structures built after World War II, from the 1950s onward, are approaching the end of their lifespans [15].

To address some of these concerns, the circular economy approach has emerged as a sustainable concept that focuses on maintaining a material’s value to the maximum extent by implementing the practices of reducing, reusing and recycling. The most sustainable method of construction is to use wastes, including both CDW waste and post-industrial waste such as ceramics [16] and plastics [17], as aggregates in concrete [18], reducing the demand for natural resources and reducing landfill pressure [19], thereby generating joint benefits for society and the environment.

The use of recycled aggregates results in concrete samples with inferior properties when compared to conventional concrete prepared using natural aggregates of gravel, sand prat or crushed rocks from quarries. Recycled aggregates are more irregular, angular and porous [20] and may have 400% greater specific area [21] and 10% less density than natural aggregates [22]. They present greater water absorption capacity levels by between 3.5% to 9.2% [23] and 70% lower abrasion resistance [24]. These factors contribute to recycled aggregate concretes having lower resistance to compression, elasticity and flexural tensile strength, together with higher water absorption and porosity, all having negative effects on the durability of the concretes [25].

Due to these detrimental characteristics, numerous investigations have sought to improve the durability of the concretes produced from recycled aggregates, using different techniques such as the removal of the attached mortar [26], pre-soaking treatment in water [27] and autogenous cleaning processes [28], among others techniques. An alternative approach involves the use of bioproducts derived from polyhydroxyalkanoate (PHA) production processes as surface treatments, which reduce water absorption, improve the quality of the surface layer and protect the concrete structure, consequently increasing its durability [29].

PHAs are a diverse family of sustainable bioplastics [30], which are accepted as potential alternatives to petrochemical plastics [31] due to their physicochemical properties [32]. PHAs are biopolyesters synthesized intracellularly by several Bacteria and Archaea as energy storage compounds [33] and are produced through the fermentation of diverse carbon sources [34] under stress conditions [35].

Currently, the cost of production of PHAs is very high [36], as pure microbial cultures require sterile conditions and the use of expensive substrates as sources of carbon, representing approximately 40% of the total production cost. Another impacting factor is the recovery and purification process required to obtain the final product, which also entails a high cost [34,37]. To reduce these production costs, the bioproducts used in the present research were derived from microbial mixed cultures (MMC), which do not require sterile conditions, combined with the utilization of low-value substrates as carbon sources [35]. The substrates considered here were pinewood bio-oil produced through fast pyrolysis of cellulose, hemicellulose and lignin and crude glycerol, an industrial residue from biodiesel production. This significant reduces the production cost [38] and the overall cost of the process while managing waste, contributing to its reduction and reuse in accordance with the circular economy approach.

In this study, the effectiveness of surface biotreatments involving bioproducts derived from PHA-producing MMCs generated from the two residual products, crude glycerol and pinewood waste, were evaluated. The aim was to increase the durability of concrete produced with recycled aggregates, preventing the penetration of water, based on the fact that water absorption causes concrete degradation by increasing the risk of frost damage, in addition to causing the transport of aggressive ions that are dissolved into the deeper pore spaces of concrete [39]. In this way, the circular economy concept could be verified in terms of both the materials used in the concrete and the treatments used to increase its durability. In addition, an application method closer to the on-site method was employed and the behavior and evolution of the biotreated concrete were evaluated in real exposure conditions.

## 2. Materials and Methods

### 2.1. Materials

#### 2.1.1. Bioproducts

The biopolymers used in this research to improve the durability of concrete were produced from microbial mixed cultures (MMC) grown with two different carbon sources, one for each bioproduct. One of them was pinewood bio-oil obtained through fast pyrolysis of waste pinewood and the second was crude glycerol obtained through biodiesel production (formed during the trans-esterification reaction). The culture medium contained tap water and ammonia as a nitrogen source. The PHA-accumulating culture enrichment was performed in a sequencing batch reactor (SBR), produced on demand from an acrylic material producer in Lisbon, with a working volume of 1500 mL [40] and operated under feast and famine conditions [35].

In these conditions, the MMC mainly produced short-chain-length PHAs, such as P(3HB) and P(3HB-3HV). Waste biomass from the reactor was suspended in water at a concentration of 2–3 g/dm^3^ to prepare two liquid bioproducts, P (MMC cultivated with pinewood bio-oil as substrate) and G (MMC cultivated with crude glycerol as substrate). Both were subject to six cycles of sonication, each with 3 min of ultrasonication (Sartorius, Goettingen, Germany) to disrupt bacterial cell membranes, followed by 3 min of rest on ice. After sonication, both bioproducts suspensions were freeze-dried to remove the aqueous phase and stored at room temperature in sealed Falcon tubes protected from the light (SP and SG, respectively, forming the sonicated pinewood bio-oil and crude glycerol bioproducts). Before use, the bioproducts were resuspended in tap water at a concentration of 2 g/dm^3^ [41].

The density of the SP was 0.997 kg/dm^3^, while that of the SG was 0.996 kg/dm^3^, measured by successive weighing of 1 mL of bioproduct on a precision balance at a controlled temperature of 23 °C. The density of both sonicated bioproducts was close to the density of water, at 0.998 kg/dm^3^ at 23 °C [42]. The kinematic viscosity (ν) levels of both bioproducts were measured in a viscometer of transparent liquids Cannon-Fenske BS 188, IP 71, (Acefe, S.A.U., Gavá, Spain) at an ambient temperature of 23 °C. The viscosity of SP was 1.2949 mm^2^/s and that of SG was 1.4460 mm^2^/s, both higher than the viscosity of water at the same temperature at 0.9333 mm^2^/s [43]. The dynamic viscosity (µ) values were very similar to the kinematic ones for both bioproducts, as shown in Table 1. The pH levels of the bioproducts, measured using a pH meter inoLab pH Level 1, (Xylem Analytics, Weilheim, Germany) at 23 °C, were slightly acidic for SP (pH = 6.36) and more acidic for SG (pH = 5.12). The colors of the bioproducts were measured using a bench-top colorimeter CR-5 (Konica Minolta model CR-5, Osaka, Japan), both in liquid state and once applied to concrete. The results are shown in Table 1, complemented by a graphical representation [44]. The intense colors of the liquid bioproduct suspensions were attenuated once applied to the concrete surfaces. One week after exposure to atmospheric conditions, the color differences between the two were nearly undetectable.

#### 2.1.2. Concrete

The composition of the concrete used in this study is shown in Table 2. Concrete components included Portland cement 42.5 N/SR CEM III/A, natural river sand (0/4 mm) and coarse aggregates (4/16 mm). The aggregate mix was a 50% blend of natural siliceous gravel and mixed recycled aggregate (MRA) from a CDW management plant operated by Áridos Valdearcos S.L, located in Valdearcos (León, Madrid, Spain). The composition of the MRA is listed in Table 3. All aggregates were characterized to ensure that they met the requirements established by the Spanish structural concrete code EHE-08 [45] and European standard EN 12620 [46]. The recycled concrete had a water/cement ratio of 0.5 and a target strength of 25 MPa, as previously evaluated by García-González et al. [47]. 

The MRA was pre-soaked to counteract the increased absorption of the water associated with the presence of bound cement mortar on the surfaces of natural aggregates or ceramic fractions. The bound cement mortar was 49% of the MRA mass and the ceramic fraction was 26%. This pre-treatment was a usual practice carried out for the manufacture of recycled concrete for applications that do not require high mechanical strength [27].

Four types of samples were prepared: six 400 × 100 × 100 mm^3^ prismatic samples, twenty five 100 × 100 × 100 mm^3^ cubic, nine 200 mm × 100 mm cylindric samples and nine 100 × 77 × 60 mm^3^ truncated cone samples. The truncated cone samples were made in plastic molds; all other samples were molded in steel molds.

The compaction of samples was carried out in three layers using a poker vibrator. Samples were flattened with a smooth steel trowel and then covered with plastic to prevent early evaporation. Specimens were stripped from their molds after 24 h, except for the ones intended for the freeze–thaw cycle test. For the last specimens, the molds were part of the setup needed to perform the test, so the samples were kept in a curing room at 20 ± 2 °C and about 100 % relative humidity for 28 days.

### 2.2. Treatment of Concrete Surface

For the water drop absorption test, ninety 50 × 50 × 100 mm^3^ specimens were cut from six 400 × 100 × 100 mm^3^ prismatic samples to compare the performance between an indoor controlled environment and outdoor exposure with natural weathering (in–out). Sixty-three specimens of the same size were cut from sixteen 100 × 100 × 100 mm^3^ cubic specimens, then divided into two groups, one used to evaluate the effects of the reapplication of bioproduct (Reap) and the other to assess the effects of the bioproduct applied on vertical exposure (V.E), for which specimens were placed with the test surface perpendicular to the ground. Another nine cubic samples of the same size were used for the determination of capillary water absorption. Nine 200 mm × 100 mm cylindric samples were used for the penetration depth of water under pressure test, and nine 100 × 77 × 60 mm^3^ truncated cone samples were used in the freeze–thaw test.

The specimens were divided into three subgroups: one-third of the specimens were treated with SG (sonicated glycerol) bioproduct; one-third were treated with the SP (sonicated pinewood bio-oil) bioproduct; one-third were treated with tap water (here designated as H_2_O), serving as the reference group. The treatments were applied after 28 days of curing on the surface that was not in contact with the mold, in an environment at 40 ± 5% relative humidity and 20 ± 2 °C.

The tested surfaces varied according to the shape of the specimens, and the bioproduct was applied at a concentration of 0.1 mL/cm^2^. The bioproduct was applied in multiple coats with a brush, so that a homogeneous distribution was guaranteed, preventing bioproduct overflow and mimicking the on-site application method. Before the application, the total amount of bioproduct was measured to minimize variability associated with brushing. 

When studying the effects of the re-application of the bioproduct, the same procedure was followed, with the second treatment applied 3 days after the first one.

Table 4 summarizes the specimens prepared for each test, the test surfaces and the amount of biopolymer applied to each of the specimens.

### 2.3. Test Method

The tests utilized to determine the effectiveness, evolution and performance of the biotreatments in protecting the surfaces of the concrete are described below.

#### 2.3.1. Hardened Concrete Resistance to Pressurized Water Penetration

The penetration depth of water under pressure test, according to EN 12390-8 [48], was conducted on the treated surfaces of the 200 × 100 specimens. Three days after applying bioproducts, the specimens were transferred to a water penetration unit, reference 270200 (Mecánica Cientifica S.A, Madrid, Spain), and subjected to 5 bars (0.5 MPa) of hydrostatic pressure for 72 h. 

The specimens were then broken via tensile strength splitting, following the guidelines given in EN 12390-6 [49], to obtain two halves perpendicular to the surface in which the water was applied, allowing the observation of the water penetration front. For this penetration front, the maximum depth reached by the water was measured, as indicated by the standard EN 12390-8 [48]. Alternatively, ImageJ software [50] was used to calculate the area of the penetration front Apf (mm^2^), which was divided by the diameter of the specimen d (mm), allowing the average penetration depth Pm (mm) to be calculated, as in Equation (1), taken from EN 12390-8 [51]:(1)$$\mathrm{Pm}=\frac{\mathrm{Apf}}{d}$$

#### 2.3.2. Water Absorption Caused by Capillarity of Hardened Concrete

Representing one of the most important parameters of concrete durability, capillary suction is one of the transport mechanisms related to the entry of harmful substances into concrete, with particularly damaging effects in areas where there are alternating climatic conditions of wetting and drying [52]. During these cyclic processes, there is absorption of water with damaging substances in solution, then when the water evaporates those substances are kept inside the concrete, increasing its concentration after each successive cycle [53]. The water penetration into the concrete results mainly from the force of the capillary absorption when the concrete is in a relatively dry state [54]. The capillary absorption behavior is closely related to the composition, pore structure and porosity of the concrete [54].

For the capillary water absorption test, the standard UNE 83982 [55] was followed. To ensure that the absorption of water only occurred in the biotreated surface, the sides of the specimens were sealed with a layer of wax measuring 1 cm thick after pre-conditioning according to UNE 83966 [56].

Three days before the end of the pre-conditioning, the surfaces of the specimens were treated with different bioproducts. The test was performed by introducing the specimens into a plastic container with a lid with a levelling plastic grille, on which the specimens were placed in contact with a 5 mm of water. The humidity and temperature in the test room were 45 ± 15% and 20 ± 2 °C according to the standard. The nine specimens were weighed at intervals of 5 min, 10 min, 15 min, 30 min, 1 h, 2 h, 3 h, 4 h, 6 h, 24 h, 48 h and 96 h until the mass was constant (considered when the difference in mass between two consecutive weighing times was smaller than 0.1%).

Additionally, the capillary absorption coefficient (k) was calculated using Equation (2), taken from UNE 83982 [55], in combination with the experimental capillary curves:(2)$$k=\frac{\delta_{a}\varepsilon }{10\sqrt{m}}$$
where *k* is the capillary absorption coefficient (kg⋅m^−2^⋅min^−0.5^), *δ_a_* is the density of the water used (1 g/cm^3^), *ε* is the concrete’s effective porosity (cm^3^/cm^3^) and *m* is the water penetration resistance caused by capillary absorption (min/cm^2^).

#### 2.3.3. Resistance of Hardened Concrete to Freeze–Thaw Cycles

The freezing and thawing cycles simulate one of the most aggressive factors affecting the concrete, causing microcracking, which in turn favors the penetration of aggressive substances into the concrete matrix, decreasing its durability [57] and possibly leading to structural collapse [58], especially in cold regions.

To perform this test, EN 1339-Annex D [59] was followed after preparation of the specimens as indicated by the standard. A plastic mold with a lid in which specimens were formed was used, together with a rubber sheet glued to the specimens, thereby ensuring the fit and perfect sealing of the specimen. This prevented leakage of the bioproducts applied three days before starting the 28 freeze–thaw cycles and of the 3% saline solution applied to the surfaces of the specimens following the guidelines set by the regulations.

For thermal insulation, a 20-mm-thick polystyrene layer with a thermal conductivity of 0.035 W/(mK), as indicated by the standard [55], was placed during the pre-conditioning of the specimens. Once this stage was completed, the specimens were transferred to a freezing chamber (−19 °C) on levelled metal grids, ensuring correct air circulation to provide the same temperature for each of the 28 cycles to which they were subjected. Periodically, for each cycle, specimens were removed from the freezing medium and left to thaw above 0 °C for 7 to 9 h, with the temperatures during the test being registered.

Given the external nature of the treatment, the material generated during testing was not removed until the end of the 28 days. After 28 cycles, the deposited material was removed from the test surface using a water spray bottle and collected in a container after being passed through a paper filter. Subsequently, the sample was placed in a stove at 50 °C, dried for 96 h and then weighed to calculate the loss of mass per unit of area (kg/m^2^) according to the regulations.

#### 2.3.4. Water Drop Absorption in Hardened Concrete

The water drop absorption test allows observations of the differences in behavior of the three types of specimens tested, i.e., the differences in behavior between the treated concrete samples kept in controlled atmosphere conditions in the laboratory (indoors) and the samples exposed to the weathering agents (outdoors), as well as their evolution over time (Figure 1a). The Reap group of specimens (Figure 1b) was used to assess the performance of concrete treated with twice the amount of bioproduct, by reapplying the bioproduct three days after the first application. Finally, the V.E. group of specimens (Figure 1c) was made to study the effects of the bioproduct on the concrete used in vertical exposure conditions, which allowed us to evaluate the behavior of treated concrete in structures perpendiculars to the ground. The absorption times were measured in the same way in all tested specimens, with the test surface parallel to the ground. The absorption rates for 9 drops of water (50 µL/drop) applied on the treated surfaces were measured, except for the reference specimens, where only 4 drops of water were applied to avoid contact with each other. The test was applied based on the RILEM procedure and schematized by Parracha et al. [60]. To accurately determine the absorption time, the difference between the exact moment in which the drop was applied until the moment of its complete absorption was calculated from video-recorded images.

##### Indoor–Outdoor

Here, 90 specimens were separated into 2 groups of 45 samples, with half of them kept in the test room at relative humidity and temperature conditions of 45 ± 15% and 20 ± 2 °C, while the other half were exposed to natural weathering (outdoors), as shown in Figure 1a. The outdoors specimens were placed on a support with holes 20 cm above the ground to avoid contact with and the accumulation of water at the base of the specimens. The placement of the specimens was vertical, with the test surfaces parallel to the ground, simulating a concrete slab, in order to reduce the risk of runoff of the bioproducts. A daily record of temperature and precipitation was kept. Each group (indoors and outdoors) was subdivided into 5 subgroups of 9 specimens (SP, SG and H_2_O in triplicates), with measurements taken within 3, 14, 28, 90 and 150 days, with day 0 considered as the day the treatments were applied.

##### Re-Application 

After the first application of the biotreatment in laboratory conditions, considering this as day 0, the 27 specimens were taken outdoors for natural weathering and placed on a 20-cm-high support with holes, as shown in Figure 1b, with the placement of the specimens in the vertical orientation and the test surface parallel to the ground. After 3 days, the specimens were applied again in the laboratory, and once the treatment had dried, were placed again outdoors with the same placement as described above. This group was subdivided into 3 subgroups of 9 specimens each (3 for each treatment), so that the evolution of the effect of the treatment could be verified by taking samples at 14, 28 and 90 days.

##### Vertical Exposure

After the first application of the biotreatment in laboratory conditions, another 36 specimens were taken outdoors for natural weathering and placed on a support of 20 cm high with holes, as shown in Figure 1c, with the test surface perpendicular to the ground (vertical exposure, V.E.). The group was subdivided into 4 groups of 9 specimens to observe the evolution of treatments after 3, 14, 28 and 90 days. For these specimens, the tests were performed 3 days after application.

Given their placement, with the test surface perpendicular to the ground, a layer of wax was used to cover the top surface, which was most exposed to atmospheric agents. In this way, absorption and modification of the sample through the untreated surface were avoided, thereby preventing this effect from interfering with the results.

## 3. Results and Discussion

When analyzing and comparing the results obtained following the bioproduct treatments, it is necessary to take into account several factors [61]: the compositions of the materials tested and their dosages; whether recycled aggregates (with or without ceramic components) were included; the type of treatment (nature of the bioproduct and its concentration); the application technique; the testing methodology [62].

### 3.1. Depth of Penetration of Water under Pressure

The pressurized water penetration test is one of the main tests used to determine the resistance to permeability [63]. Therefore, it is an important parameter for determining the durability of the hardened concrete.

Figure 2a shows the maximum values achieved by the penetration front of the different specimens treated with the two bioproducts SP and SG and with H_2_O (control). Recycled aggregate concrete control specimens (H_2_O-treated concrete) had a penetration depth of 30 mm, confirming that the pore structure was sufficiently impermeable to guarantee durability, as stipulated in Spain’s structural concrete code [45] and corroborated by Cantero et al. [64]. The value was in fact lower than those obtained by other authors, such as Martínez-Large et al. [65], who observed maximum depths of penetration of between 34 and 50 mm, dependent on the type of mixture with the same percentages of recycled aggregate replacement.

The application of both bioproducts resulted in a decrease in the penetration of water into specimens. In the case of SP, the reduction of the maximum penetration values was in the order of 25%, whereas a smaller reduction of 4% was detected for SG (Figure 2a). However, the effect of the treatment with both bioproducts can be better seen in Figure 2b, which shows the mean depth of penetration (Pm) value for the specimens. Due to the irregular area enclosed in the penetration front, which was present in all specimens, the mean depth value offered a more representative value of the protective effect that the bioproducts had on the concrete relative to the penetration of water under pressure. The differences between the values of H_2_O-treated concrete and biotreated concrete were considerably greater, in the order of 51% and 42% for SP and SG, respectively. These values were higher than those obtained by Husni et al. [66], with an 18% difference between specimens coated with superhydrophobic rice husk ash and uncoated concrete and were similar to those found by Serrano-González et al. [67] when applying MMC-derived bioproducts with pinewood bio-oil and crude glycerol as carbon sources.

Considering both the maximum and mean values (Figure 2), the greater effectiveness in protecting the surface of the hardened concrete from the penetration of water under pressure was shown by the SP treatment. The water penetration front in SP-treated specimens was also more uniform (with a more uniform distribution of the front), which resulted in a more homogeneous profile and a lower penetration surface, as seen in Figure 3. This difference in distribution was most likely due to the difference in viscosity between the two bioproducts. The lower viscosity of the SP might have allowed a better diffusion throughout the concrete surface, generating a more consistent protective film against the penetration of water under pressure.

### 3.2. Water Absorption caused by Capillarity 

Figure 4 shows the capillary absorption curves of the concrete samples with recycled aggregate treated with either the bioproducts (SP, SG) or water (in the case of control samples). The curves can be divided into two phases, with the first one related to the filling of water through the finer capillary pores, where the action of capillary forces is greater than the gravitational forces [68,69]; and the second phase corresponding to the continuity of the filling through the air pores caused by the diffusion and dissolution of air [70,71]. The results showed that the amount of water absorbed increased with the time of contact with water, reaching a constant mass after 96 h. Both bioproducts decreased the amount of water absorbed, by more than 17% for SP and by almost 8% for SG compared to water-treated specimens. These results were very similar to those obtained when using a MMC–glycerol-based concrete treatment [29], with improvements of 20% with SG and 13% with NSG (non-sonicated glycerol) and less than the 40% reduction, as found by Chandra et al. [72], using cactus extract impregnated in concrete. Moreover, the results obtained with the SP treatment were similar to the ones reported by Oliveira et al. [41] when MMC cultivated with the same carbon substrate was applied in air lime mortars, obtaining a reduction of 17% in water absorption. However, the results achieved using the present treatments were superior to those obtained by Molina et al. [73] when studying the effects of cactus mucilage within air lime mortar, which resulted in a reduction of 5%.

The decrease in water absorption due to the capillarity of the biotreated specimens compared to control specimens may be explained by the fact that the bioproducts acted on the larger pores walls, making it difficult for the pores to transport water, in accordance with Scarfato et al. [68], Xu et al. [74] and Chen et al. [39]. However, due to the high molecular weights of the bioproducts, the smaller pores were not affected. These results agree with the results of the pressurized water test, corroborating the fact that the protective capacity of SP was greater than that of SG. The explanation for this efficacy may follow the same reasoning, caused by the difference in viscosity between the two bioproducts, resulting in the more homogeneous distribution.

The efficacy of the hydrophobic agents generated from MMC applied on the concrete surface was also corroborated by the mean values of capillary absorption coefficients (k) shown in Table 5. The values obtained fulfilled the minimum requirement for the coated concrete ingress established in EN 1504-02 [75], whereby the water permeability coefficient should not exceed 1.29 × 10^−2^ kg⋅m^−2^⋅min^−0.5^, showing that the water proofing capacity of the bioproducts reduced the absorption coefficient in the case of SP by more than 18% and in the case of SG by 8.5% when compared to the reference specimens (H_2_O).

### 3.3. Resistance to Freeze–Thaw Cycles

The damage generated by freezing and thawing cycles can be explained by three factors: hydraulic pressure due to volume increase when ice forms [76]; osmotic pressure generated in the pore system caused by the movement of liquid water towards pores containing ice to restore the thermodynamic equilibrium [77]; the pressure induced by the growth of crystals in pores and their interactions with pore walls [78]. Another major factor that affects the resistance of concrete to freeze and thaw cycles is its composition, in particular the presence of recycled aggregates, since they present more freeze–thaw resistance than natural aggregate concrete due to the lower scaling rate and narrower cracks [79].

Figure 5 shows the weight loss caused by freeze–thaw cycles. The surface loss of cement and the removal of aggregates are some of the typical effects of frost damage, which lead to mass loss of the specimens. After the 28 freeze–thaw cycles, the appearance of numerous cracks, the progressive elimination of small chips in the concrete surface and scales in the aggregates (in the case of ceramic aggregates) were recorded. These damages led to the exposure of aggregates and even the formation of large holes due to the loss of aggregates, with the latter only observed in the control (treated with H_2_O) specimens.

Regarding the biotreated specimens, the application of the bioproducts slightly improved the behavior of the recycled concrete subjected to the freeze–thaw cycles. Less mass was lost relative to the control samples (10.5% less for the SP-treated specimens and almost 9% for the SG specimens). As mentioned, the appearance of holes due to the loss of aggregates only occurred in the control specimens. These subtle improvements obtained with the bioproducts may be due to the formation of a protective layer that is resistant to water ingress through the surface cracks of the recycled concrete, decreasing water penetration. Therefore, the effects generated in each of the cycles will have a smaller impact with lower mass loss in the treated specimens. The mass loss results were similar to the results for coarse recycled aggregate concrete obtained by Luan et al. [79] after 28 cycles. Still, they were much lower than those recorded by Wiktor and Jonkers [80] for conventional concrete samples treated with a bacteria-based repair system, where both the biotreated and control samples showed mass loss values higher than 1 kg/m^2^ after 7 cycles.

Figure 6 shows the superficial damage for the three different types of specimens tested. The damage in the specimens was greater than that described by Chandra et al. [72] in mortar specimens containing cactus extract. Additionally, the mass loss in this study was greater than the loss for plant concrete samples studied by Ahmad et al. [81], although in their case the concrete types were very different and the number of freeze–thaw cycles was limited to 20.

### 3.4. Water Drop Absorption 

All of the results from previous tests support the existence of a water-resistant effect produced by the first application of both SP and SG biotreatments, suggesting a slight waterproofing effect of these treatments. The results of the water droplet absorption tests are shown in Figure 7 (comparison of the indoor–outdoor effect) and Figure 8 (re-application, vertical exposure).

#### 3.4.1. Indoor–Outdoor Effect

Figure 7a shows the results for specimens tested in environments with controlled temperature and relative humidity (indoor) and with natural weathering (outdoor) conditions. In both environments, the two bioproducts significantly increased the absorption time relative to the reference. After 3 days of application, the outdoor tested samples treated with SP increased their absorption by more than 49 times and those treated with SG increased by more than 58 times. In the case of indoor specimens, both treatments increased their absorption by over 88 times compared to the absorption time of the reference. The results for both conditions were in line with the ones obtained by García-González et al. [82] when applying the two MMC bioproducts made with pinewood bio-oil and crude glycerol as substrates on recycled concrete. The results obtained in the present work were much higher than those reported by Parracha et al. [60] when utilizing iron-based bioproducts on clay plastering mortars. For these authors, the most effective treatment at 4 days resulted in absorption times approximately 18 times higher than those of the control. They were also higher than the results obtained by García-González et al. [83], with an improvement of 12 times over the control, when applying MMC–glycerol-based treatments on cement mortar. For the same type of bioproduct using MMC–glycerol on air lime bioformulated mortars, Oliveira et al. [41] recorded a delay time of approximately 7 times greater than the control. It should not be forgotten that the properties observed for a surface treatment depend on the characteristics of the substrate and on the concentration of the bioproduct [84].

The efficacy of both biotreatments declined progressively, being most evident in outdoor specimens exposed to climatic agents (Figure 7a), in particular due to the high rainfall recorded in the first 28 cycles (Figure 7b), which caused an important washing effect on the bioproduct. However, after 150 days of treatment, the outdoor exposed test samples continued to perform better when compared to the reference—more than 19 times better in the case of SG and more than 39 times better in the case of SP. Indoor specimens showed a reduced loss of efficiency because they were protected from atmospheric agents, and after 150 days of treatment specimens treated with SG exceeded the absorption time of the control by more than 77 and SP by more than 78 times. It was also possible to observe the difference in behavior between the two bioproducts in the outdoor tested concrete samples. SP showed greater protection of the concrete surface from water absorption, in accordance with the results obtained by García-González et al. [82] in indoor conditions, with less loss of efficiency over time. This loss of efficacy is common in superficial organic treatments [85,86], although the durability of the surface treatments is influenced by the temperature cycle, dry–wet cycle and radiation [87,88]. The increased uptake time of SG in the first trial, which also coincided with the reference behavior, was related to the high precipitation (Figure 7b) in the days prior to the test. In the case of the indoor specimens, the behavior of both biopolymers was very similar, while the differences did not exceed 10% in any of the measurements.

#### 3.4.2. Reapplied Specimens (Reap)

Specimens treated twice with both bioproducts presented significantly improved absorption times compared to the reference specimens, increasing by more than 130 times in the case of SG and exceeding the absorption time by more than 150 times in the case of SP (Figure 8a).

The differentiating element observed in this re-application assay was greater in terms of absorption times. Comparing the outdoor assays with a single application and re-application (Figure 8a,b) for the same test days (14 and 28 days), the decrease observed for SG was only of 8% while for SP it was 16%, as opposed to 70% and 63%, respectively, in the single-application assay. The control specimens decreased in absorption time by 20% in the re-application assay. Under this new situation, the bioproducts presented a small decrease in their protective capacity, despite being specially subjected to two atmospheric agents that have great impacts on the durability of surface treatments, such as high temperatures and ultraviolet radiation (UV) [89]. This assay was started in July, meaning specimens were exposed during the following months to high values of these two factors [90]. Some researchers were able to demonstrate the great influence that high temperatures [91,92] and exposure to UV [93] have on surface hydrophobic treatments applied to concrete. These facts reinforce the positive influence of the bioproduct re-application and is an indicator of its effectiveness. The greater protective capacity of SP versus SG was maintained, with values often differing by more than 24%, despite sometimes matching.

#### 3.4.3. Vertical Exposure (V.E.) 

Figure 8b shows the results for specimens in outdoor conditions, which were tested using the water drop absorption method to assess the effects of biopolymers on vertical exposure (with the test surface perpendicular to the ground). The specimens treated with the bioproducts presented significant increases in the absorption times of the water droplets, being 75 times longer in the case of SP and 47 times longer in the case of SG as compared with the reference, for the first measurement after 3 days of treatment. After 90 days, those values were 37 times and 33 times longer than the reference time for SG and SP, respectively.

Another clear observation that can be made from Figure 8b is the decrease in the efficacy of the bioproducts over time, reducing the absorption time for SP by 75% and for SG by 65% at 90 days. An aspect that must be taken carefully into account is that these treatments have low penetrability by nature [62], meaning the exposure of the tested surfaces may lead to drip-off of the bioproducts. Again, the greater protective capacity of SP over SG was demonstrated, with differences between both bioproducts of more than 31%.

This may have been due to the fact that SP had a more homogeneous distribution than SG over the treated surfaces, as can be seen in SEM images (Figure 9) [29,67] taken after the assays. The better distribution of the SP biotreatment may have been due to its lower viscosity. In addition, these results may also relate to the higher degree of acidity presented by SG, although its contribution to the results was very weak [94].

Some researchers [95,96,97,98] have studied how the effects of surface treatments evolve in natural environments. A comparison of results is somewhat difficult given the variability in the exposure conditions, the different types of concrete treated and tested and the different types of treatments applied. Seneviratne et al. [95] investigated the evolution of three organic surface coatings applied in vertical exposure conditions and concluded that the protective effect decreased over time, mostly due to the decreased elasticity and viscosity caused by temperature and time. This was corroborated by Pan et al. [89] and is also in accordance with the results obtained in this research. Li et al. [98] proved that the exposure of 4 surface treatments to accelerated aging under outdoor natural conditions accelerated the aging of the organic film coatings.

## 4. Conclusions

Two bioproducts obtained from sonicated waste biomass of PHA-accumulating MMC, using crude glycerol (SG) and pinewood bio-oil (SP) as the carbon sources, were applied on the surfaces of concrete samples produced with partially recycled aggregate. The protective effects of the bioproducts were evaluated experimentally through six different tests. The conclusions obtained were as follows:The treatments with the bioproducts showed significant improvements relative to the reference concrete (treated with H_2_O) in all tests, although the improvements were particularly noticeable in the pressurized water penetration and water drop absorption tests;The effects of the bioproducts in the water under pressure test were better perceived via the average penetration values due to the irregularity of the profiles, showing the greater uniformity of the SP due to the better distribution of the bioproducts;The bioproducts act on the larger pores walls, making water transport difficult, although due to their high molecular weight they do not seem to interfere with the smallest pores;The protective layer of the bioproducts prevents the entry of water through the cracks, explaining the smaller loss of mass in the freeze–thaw test relative to the control and the non-appearance of holes due to the loss of aggregates. The efficacy of the biotreatments was reduced with time under natural exposure conditions, although after 150 days the decrease in water absorption was still 19 times longer in the case of SG and 39 times longer for SP when compared to reference specimens;The re-application of bioproducts after 3 days resulted in a significant improvement in the treatment efficacy, which declined over time in natural weathering conditions but remained mostly constant after 90 days;The use of both bioproducts in a vertical exposure test generated a significant protective effect, although their efficiency decreased more than 60% after 90 days.

The incorporation of two waste materials such as crude glycerol and pinewood bio-oil to produce the bioproducts used to improve the capacity and durability of recycled concrete has a positive environmental impact. This strategy involves two routes leading to a more circular economy, both by reducing waste and improving waste management, thereby reducing the consumption of energy and natural resources.

In summary, this research shows the effectiveness of both bioproducts in protecting the surfaces of recycled aggregate concrete samples against water penetration and the consequent risk of deterioration and decrease in their useful lifespan; that is, the useful lifespan and performance of recycled concrete are increased, encouraging the use of construction wastes as aggregates, thereby reducing the volume of such waste.

## Figures and Tables

**Figure 1 materials-15-01181-f001:**
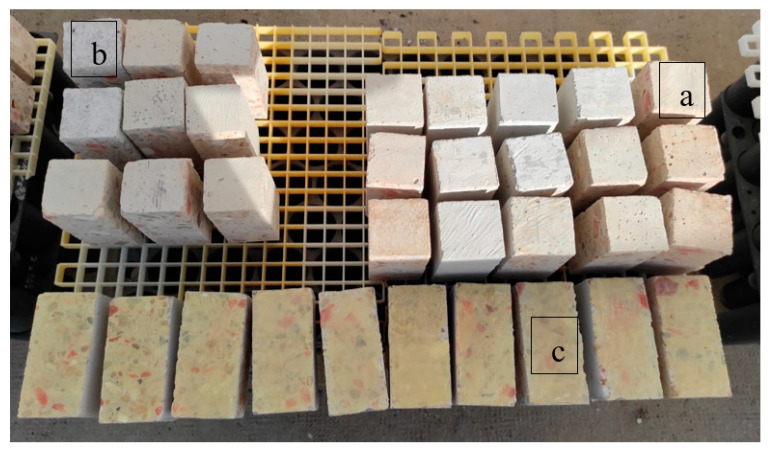
The water drop absorption test specimens: (**a**) outdoor specimens from the test performed to compare indoor and outdoor treatments; (**b**) specimens used to analyze the effect of the reapplication of treatments; (**c**) specimens used to evaluate treatment effects on vertical exposure.

**Figure 2 materials-15-01181-f002:**
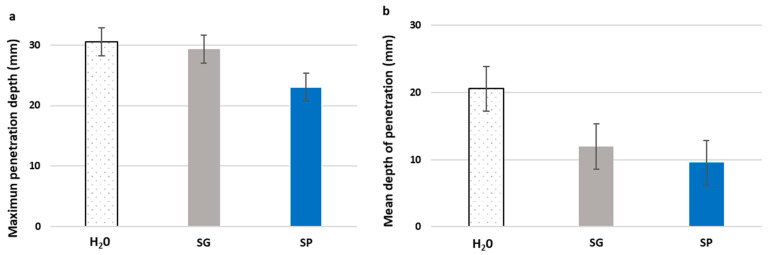
Maximum penetration depth (**a**) and mean depth of penetration (**b**) values for concrete specimens treated with bioproducts made using pine bio-oil (SP) and crude glycerol (SG) and with water (H_2_O). Error bars indicate standard error values.

**Figure 3 materials-15-01181-f003:**
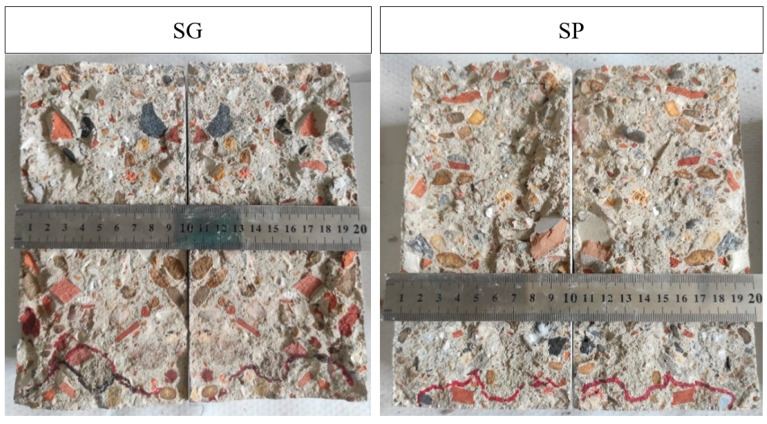
Examples of penetration fronts following water under pressure testing in MMC–glycerol-based (SG) and pine-bio-oil-based (SP) treated concretes.

**Figure 4 materials-15-01181-f004:**
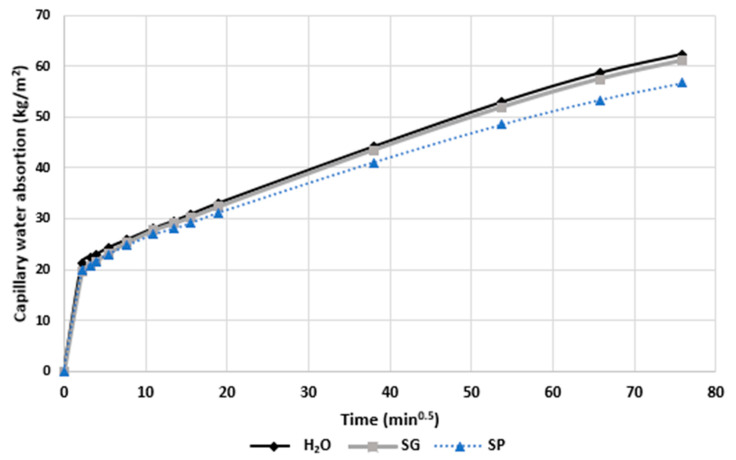
Capillary curves for concrete specimens with the surfaces treated with water and MMC feed made from crude glycerol (SG) and pine bio-oil (SP) bioproducts.

**Figure 5 materials-15-01181-f005:**
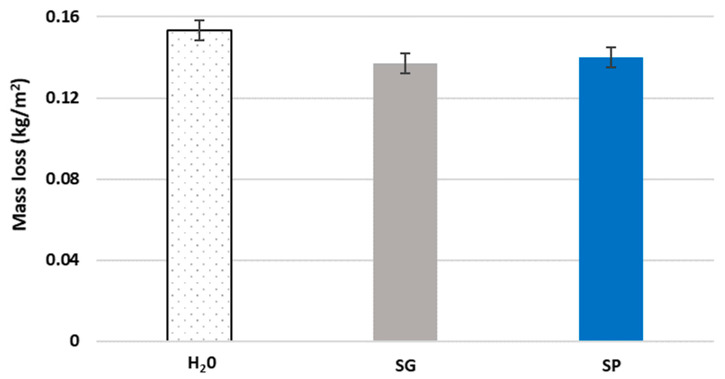
Average mass loss effects caused by freeze–thaw cycles in concrete samples treated with water (H_2_O) and bioproducts made using pine bio-oil (SP) and crude glycerol (SG). Error bars indicate standard error.

**Figure 6 materials-15-01181-f006:**
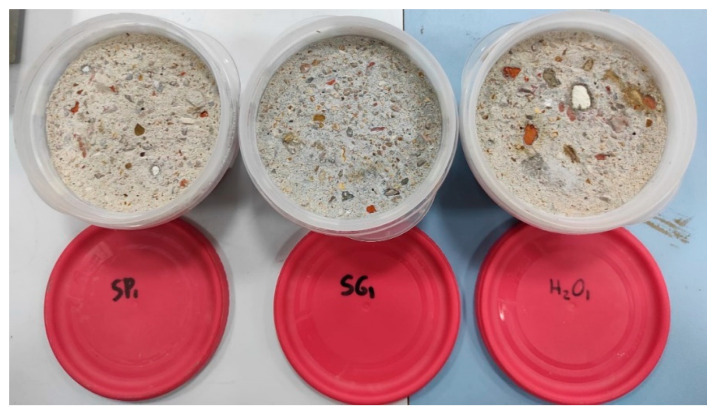
Surfaces of control, SP and SG specimens after the freeze–thaw test.

**Figure 7 materials-15-01181-f007:**
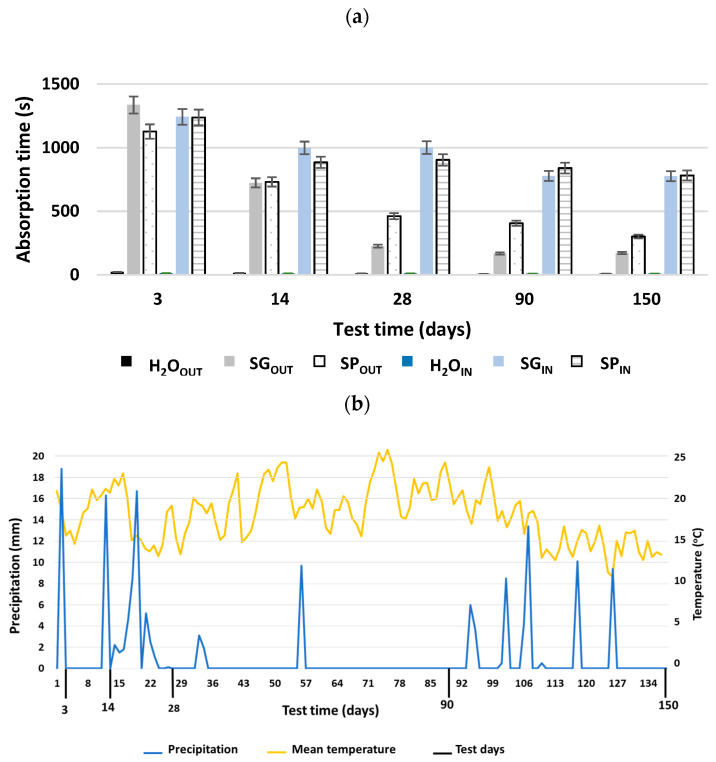
Mean water drop absorption times for outdoor and indoor exposed tested concrete samples treated with SP, SG and water (**a**). Records of precipitation and average temperature during natural weathering of concrete samples (**b**).

**Figure 8 materials-15-01181-f008:**
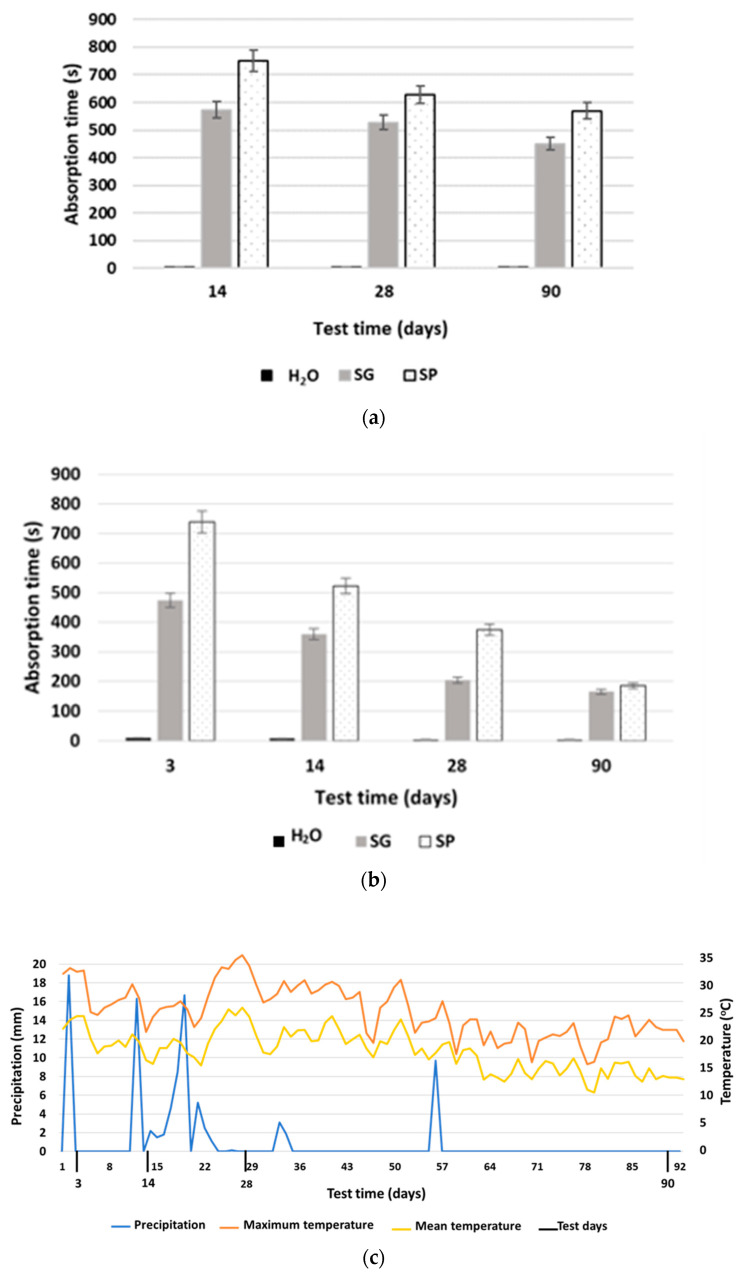
Mean water drop absorption times for Reap specimens (**a**) and V.E. specimens (**b**) treated with SP, SG and water. Records of maximum and average temperatures and precipitation during the test (**c**).

**Figure 9 materials-15-01181-f009:**
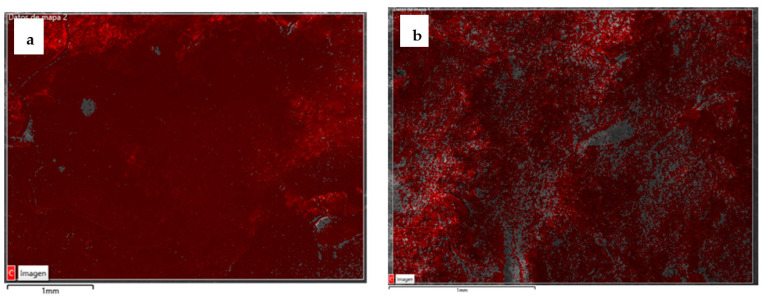
Secondary electron image captured using SEM, caused by layers superimposed by the carbon distributions of SP (**a**) [67] and SG (**b**) [29].

**Table 1 materials-15-01181-t001:** Characterization of bioproducts in terms of density (ρ), kinematic viscosity (ν), dynamic viscosity (µ), pH and colorimetry (L*a*b space coordinates and graphic representation of colors).

	ρ	ν	µ	pH	Condition	Colorimetry
	(kg/dm^3)^	(mm^2^/s)	(Pa·s)			L	a	b	
**SP**	0.99722	1.29491	1.29132	6.36	liquid	15.675	21.2	26.53	
indoor	61.846	5.658	19.468	
outdoor	73.142	1.366	10.486	
**SG**	0.996	1.44603	1.44025	5.12	liquid	28.18	7.48	28.79	
indoor	68.822	3.076	14.53	
outdoor	73.696	1.844	12.246	

**Table 2 materials-15-01181-t002:** Recycled concrete composition per m^3^.

Material	Composition (/m^3^)
Water (L)	215
Cement (kg)	391
Sand (kg)	716
Gravel (kg)	447
MRA (kg)	447

**Table 3 materials-15-01181-t003:** Composition of mixed recycled aggregates (MRA).

Components	wt%
Unbound aggregate (natural aggregate with no attached cement mortar)	23.98
Masonry and fired clay (bricks, tiles, stoneware, sanitary ware, …)	26.18
Concrete and mortar (natural aggregate with bound cement mortar)	49.38
Asphalt	0
Glass	0.33
Gypsum	0
Other impurities (wood, paper, metals, plastic, …)	0.06

**Table 4 materials-15-01181-t004:** Treated specimens and tests.

Test		Specimens		Test Surface (mm^2^)	Volume of Bioproduct per Specimen (mL)
		N^o^	Dimensions (mm^3^)		
Water under pressure		9	200 × 100	π × 50^2^	7.85
Capillary		9	100 × 100 × 100	100 × 100	10.0
Freeze–thaw		9	100 × 77 × 60	Π × 50^2^	7.85
Water drop absorption	In-out	90	50 × 100 × 100	50 × 50	2.50
	Reap	27	50 × 100 × 100	50 × 50	5.00
	V.E.	36	50 × 100 × 100	50 × 50	2.50

In–out—indoor–outdoor; Reap—reapplication; V.E.—vertical exposure.

**Table 5 materials-15-01181-t005:** Capillary water absorption coefficient of concrete specimens made with recycled aggregate treated with water, SG and SP.

Treatment	Capillary Water Absorption Coefficient (kg⋅m^−^^2^⋅min^−^^0.5^)	Standard Deviation
H_2_O	8.63 × 10^−^^4^	7.32 × 10^−^^5^
SG	7.89 × 10^−^^4^	2.77 × 10^−^^5^
SP	7.06 × 10^−^^4^	7.31 × 10^−^^5^

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
