# Peer review of "Biotreatments Using Microbial Mixed Cultures with Crude Glycerol and Waste Pinewood as Carbon Sources: Influence of Application on the Durability of Recycled Concrete"

_materials, 2022, doi:10.3390/ma15031181_

Round 1
Reviewer 1 Report
The incorporation of two waste materials such as crude glycerol and pinewood biooil to produce the bioproducts used to improve the capacities of recycled concrete and its durability have a positive environmental impact. The author's research can provide a new research idea for improving the durability of recycled aggregate concrete. This will improve the utilization rate of recycled aggregate concrete and effectively improve the service life of recycled aggregate concrete.
- When the author carries out different experiments, the number of test cycles is different, why?
- Figure 4: The interpretation of the difference of capillary curves of concrete with the surface treated with water and MMC based in crude glycerol (SG) and in pine bio-oil (SP) needs to be strengthened.
- In section 3.4.1, the author compares the research results with previous studies, and the obtained results were much higher than previous studies, is it mainly because of time, please explain.
- Is section 3.4.3, is it appropriate for the author to analyze the results of Figure 7 here?
- The title should be revised because of the wide range of durability.
- The conclusion should include the all findings of studies.
Author Response
Response to Reviewer 1
When the author carries out different experiments, the number of test cycles is different, why?
Because these are different and complementary tests, although with different objectives. Therefore, those tests do not follow a specific regulation, and have been adapted and designed according to the aim of the analysis, also based on results previously obtained.
Figure 4: The interpretation of the difference of capillary curves of concrete with the surface treated with water and MMC based in crude glycerol (SG) and in pine bio-oil (SP) needs to be strengthened.
Thank you for the recommendation. To complement the information the text has been modified.
The results showed that the amount of water absorbed increased with the time of contact with water, reaching a constant mass after 96 h. Both bioproducts decreased the amount of water absorbed more than 17 % for SP and almost 8 % for SG relative to the water treated specimens.
In section 3.4.1, the author compares the research results with previous studies, and the obtained results were much higher than previous studies, is it mainly because of time, please explain.
Thanks you for the comment.
The results obtained in the present work were compared with the ones reported in the literature. One should take into consideration that the type of treatments are different, bioproducts were applied differently to distinct construction materials. Nevertheless, the results obtained are in line with those obtained by García-González et al (83), using the same type of bioproducts, produced in similar growth conditions.
Is section 3.4.3, is it appropriate for the author to analyze the results of Figure 7 here?
Thank you for the advice, there was a typo in the text that has been corrected accordingly.
The title should be revised because of the wide range of durability.
The title was revised and was modified to: Biotreatments using microbial mixed cultures with crude glycerol and waste pinewood as carbon sources: influence of application on the durability of recycled concrete
The conclusion should include the all findings of studies.
Thank you for the comments. Conclusions have been modified:
- The treatments with the bioproducts showed a significant improvement relative to the reference concrete (treated with H2O), in all the tests, particularly noticeable in the pressurized water penetration and water drop absorption tests.
- The effect of the bioproducts in the water under pressure test is better perceived with the average penetration values due to the irregularity of the profiles, pre-setting a greater uniformity of the SP due to a better distribution of the biopolymer.
- The bioproducts act on the larger pores walls hampering the transport of water, but due to their high molecular weight they do not seem to interfere in the smallest pores.
- The effect of the protective layer of bioproducts, prevents the entry of water through the cracks, explaining a smaller loss of mass in the ice-thaw test relative to the control, and the non-appearance of holes due to the loss of aggregates. The efficacy of the biotreatments reduces with time under natural exposure conditions, although after 150 days the decrease in water absorption was still 19 times longer in the case of SG and 39 times for the SP, when compared to reference specimens.
- The re-application of bioproducts after 3 days resulted in a significant improvement in the treatment efficacy, that reduced over time in natural weathering conditions, remaining mostly constant after 90 days.
- The use of both bioproducts tested in a vertical exposure generated important protective effect, although their efficiency decreased more than 60 % after 90 days.
Reviewer 2 Report
In this article, crude glycerol and pine bio-oil were used as carbon sources for mixed culture of microorganisms to produce polyhydroxyalkanoates, and the durability performance of recycled aggregate concrete was analyzed and evaluated. Tests such as water penetration test under pressure, capillary absorption test, and freeze-thaw cycle test were conducted, and the results of each test were significantly improved compared to the control group, effectively extending the life of the recycled aggregate concrete. This paper is innovative and rich in experiments. However, there are some minor problems in the article, and I hope the authors can revise them seriously. The specific problems are as follows.
- It is mentioned in the article that SP spreads better on the concrete surface and forms a dense protective film, can microscopic tests be added to support or analyze this, please ask the authors to analyze and explain.
- In the analysis of freeze-thaw cycle test, the authors mentioned that the surface loss rate of SP-treated samples was reduced by 10.5%, while that of SG-treated samples was almost reduced by 9%, which is inconsistent with the conclusion that the protective film formed on the surface of SP is more dense, please explain accordingly.
- In section 3.2 of the article, it is mentioned that the water absorption rate of SP and SG test groups decreased by 17% and 8% respectively with the increase of time.
- There are several errors in the article, it is suggested to check the content of the article in detail, for example, what does the "table 0" mentioned in line 137 refer to?
- There are many discrepancies between the graphical representations in the article, such as whether Figure 8 should be the content of Section 3.4.3, and whether Figure 7a and b are placed in the opposite order?
- Does the fact that the control group itself was treated with water during the tests of water droplet absorption in the article have any effect on the final test results? Please provide an explanation for this.
- In part of the article, the biologics were recoated after 3 d. What is the significance of recoating? We all know that recoating is equivalent to increasing the concentration and the thickness of the biologics, so the relevant properties of the sample must be improved? Could the authors please give some explanation.
Reviewer 3 Report
Dear Author,
The present research is on Durability improvement of recycled concrete using microbial mixed cultures with crude glycerol and waste pinewood as carbon sources, as very few people are working on this title, and the novelty of the research is coming through its title of the paper, and author well documented the present research using microbial cultures with crude glycerol and waste pinewood as carbon source nearabout no one used the same and reflect its novelty of the paper.
The authors are showing the durability using two carbon sources and the results were found good. The evidence is also enough and validates the results nicely through his characterization and analysis study in graphs. Also showing the capillary rise coefficient table, freezing and thawing cycles study, weight loss, the surface of control, water drop absorption, gives insight nicely and outcome of this research can be used by large communities as the topic of research is having massive interest for the main objective of this paper. Application in field study and pilot study is the weakness and that to be explored further.
Line no. 555 to 564 giving other citations of the work and improving further the discussion in the same paragraph.
Line no. 328 and 329 show the test to be conducted after 3 days in vertical exposure.
The present paper shows the effectiveness of both bioproducts to protect the surface of recycled aggregate concrete against water penetration and the consequent risk of deterioration and decrease of their useful life. This leads to increasing the useful life and performance of recycled concrete, encouraging the use of construction waste as aggregates, thereby contributing to reducing the volume of such waste. Paper can be accepted for the esteemed journal.
Reviewer 4 Report
Please see the file attached.

Author Response
Answer to Reviewer 4
Introduction – The authors should mention that apart from the use of CDW as a recycling aggregate, post-industrial waste, e.g. ceramics, plastics, etc. are also successfully used on a large scale. Examples of references in this topic:
Thank you for the comment. These references have been included in the introduction.
The main way to make construction sustainable is to use waste, both CDW waste and post-industrial waste, such as ceramics [17] and plastics [18], as aggregates in concrete [19], reducing the demand for natural resources and releasing the landfill pressure [20], and therefore generating a joint benefit for the society and the environment.
Line 136 – There is something wrong with this sentence, as if the authors had omitted some information
Thank you for the advice. This mistake has been corrected in the manuscript.
The density of the SP was 0.997 kg/dm3 while that of the SG was 0.996 kg/dm3, measured by successive weighing of 1 mL of the bioproduct suspension on a precision balance, at a controlled temperature of 23 °C.
Table 3 – How were such accurate results for MRA compositions obtained?
Using EN 933-11 (2009)-Tests to determine the geometric properties of aggregates. Part 11: Classification test of coarse aggregate components.
Initially the maximum size of the aggregate is determined and based on this, the amount of the sample to be analysed was determined.
It is then sifted and washed, the number of types of materials in the sample is classified according with the standard and the percentage of each of them is determined.
Reviewer 5 Report
In this study, the durability of concrete specimens made from recycled aggregate has been improved using bioproducts of polyhydroxyalkanoates-accumulating microbial mixed cultures. The topic covered in this study is interesting considering the use of recycled aggregates and biowaste in construction with a circular economy approach. The manuscript merits to be published in a form of a journal article in this reviewer's opinion, provided the following comments are addressed:
Line 75-76: These values are probably in comparison with those of natural aggregates. If so, please indicate the comparison reference.
Line 85: If available, can you provide more references here?
Line 136 Table 0 is probably a typo.
Line 136: Please rephrase this sentence.
Line 227-229: Please consider using the indexes in a subscripted format.
Line 362: Same as above.
Line 373: The quality of the presentation of the results can be improved in figure 2.
Line 444: Same as above.
Line 536: Has the abbreviated form V.E. been introduced earlier in the text?
Line 567: Same for SG and SP.
Author Response
Answer to Reviewer 5
Line 75-76: These values are probably in comparison with those of natural aggregates. If so, please indicate the comparison reference.
Thank you for the comment. It has been included in the manuscript.
The use of recycled aggregates results in concrete with inferior properties when compared to conventional concrete made with natural aggregates from gravel or sand prat or crushed rocks from quarries.
Line 85: If available, can you provide more references here?
Thank you for the recommendation, more references have been included in the manuscript
Due to these detrimental characteristics, numerous investigations have sought to improve the durability of the concrete produced with recycled aggregates, using different techniques such as removal of attached mortar [27], pre-soaking treatment in water [28], autogenous cleaning process [29], among others techniques. An alternative approach focus on the use of bioproducts derived from polyhydroxyalkanoates (PHA) production processes as surface treatment, that reduces water absorption, improves the quality of surface layer and protects the concrete structure, consequently increasing its durability [30].
Line 136 Table 0 is probably a typo. Line 136: Please rephrase this sentence.
Thank you for the comment. The mistake has been corrected.
The density of the SP was 0.997 kg/dm3 while that of the SG was 0.996 kg/dm3, measured by successive weighing of 1 mL of the bioproduct suspension on a precision balance, at a controlled temperature of 23 °C.
Line 227-229: Please consider using the indexes in a subscripted format.
Thank you for the comment. It has been corrected accordingly.
Line 362: Same as above.
It has been corrected accordingly.
Line 373: The quality of the presentation of the results can be improved in figure 2.
Thank you for the suggestion. The figure was improved.
Line 444: Same as above.
Thank you for the recommendation. The figure was improved.
Line 536: Has the abbreviated form V.E. been introduced earlier in the text? Line 567: Same for SG and SP.
Thank you for this comment. both V.E., SG and SP are introduced on more than one occasion in the text.